# MambaByte: Token-free Selective State Space Model

**Junxiong Wang, Tushaar Gangavarapu, Jing Nathan Yan, Alexander M. Rush**
Cornell University
{jw2544,tg352,jy858,arush}@cornell.edu

## Abstract

Token-free language models learn directly from raw bytes and remove the inductive bias of subword tokenization. Operating on bytes, however, results in significantly longer sequences. In this setting, standard autoregressive Transformers scale poorly as the effective memory required grows with sequence length. The recent Mamba state space model (SSM) development offers an appealing alternative approach with a fixed-sized memory state and efficient decoding. We propose MambaByte, a token-free adaptation of the Mamba SSM trained autoregressively on byte sequences. In terms of modeling, we show MambaByte to be competitive with, and even to outperform, state-of-the-art subword Transformers on language modeling tasks while maintaining the benefits of token-free language models, such as robustness to noise. In terms of efficiency, we develop an adaptation of speculative decoding with tokenized drafting and byte-level verification. This results in a $2.6\times$ inference speedup to the standard MambaByte implementation, showing similar decoding efficiency as the subword Mamba. These findings establish the viability of SSMs in enabling token-free language modeling.

## 1 Introduction

When defining a language model, a base tokenization is typically used—either words (Bengio et al., 2000), subwords (Schuster & Nakajima, 2012; Sennrich et al., 2015; Wu et al., 2016; Wang et al., 2020), or characters (Gao et al., 2020b). Of these, subword tokenization has been the most popular choice, as it achieves a natural compromise between training efficiency and the ability to handle out-of-vocabulary words. However, several works, e.g., Xue et al. (2022), have noted issues with subword tokenizers, such as a lack of robustness to typos, spelling and capitalization variations, and morphological changes.

Modeling byte sequences, i.e., mapping from raw data to predictions without any intermediate tokenization, offers an alternative approach with less inductive bias (Choe et al., 2019; Al-Rfou et al., 2019; Clark et al., 2022; Tay et al., 2022; Xue et al., 2022; Yu et al., 2023). Compared to subword models, byte-level language models can generalize more easily across orthographic and morphological variants. Of course, modeling text as bytes means that the resultant sequences are significantly longer than their subword counterparts. This change pushes the modeling and efficiency issues upstream into the architecture itself.

These issues are particularly pronounced for autoregressive Transformers (Vaswani et al., 2017), which dominate language modeling (Brown et al., 2020; Touvron et al., 2023). Due to the quadratic nature of attention, Transformer efficiency scales poorly for long (byte) sequences (Zhang et al., 2022). Researchers have *compressed* the internal Transformer representation to work with long sequences, for instance, developing length-aware modeling approaches (Dai et al., 2020; Nawrot et al., 2022), where groups of tokens are merged within the intermediate layers. The MegaByte Transformer (Yu et al., 2023) is of particular relevance, which uses compression in the form of fixed-size patches of bytes as a subword analog

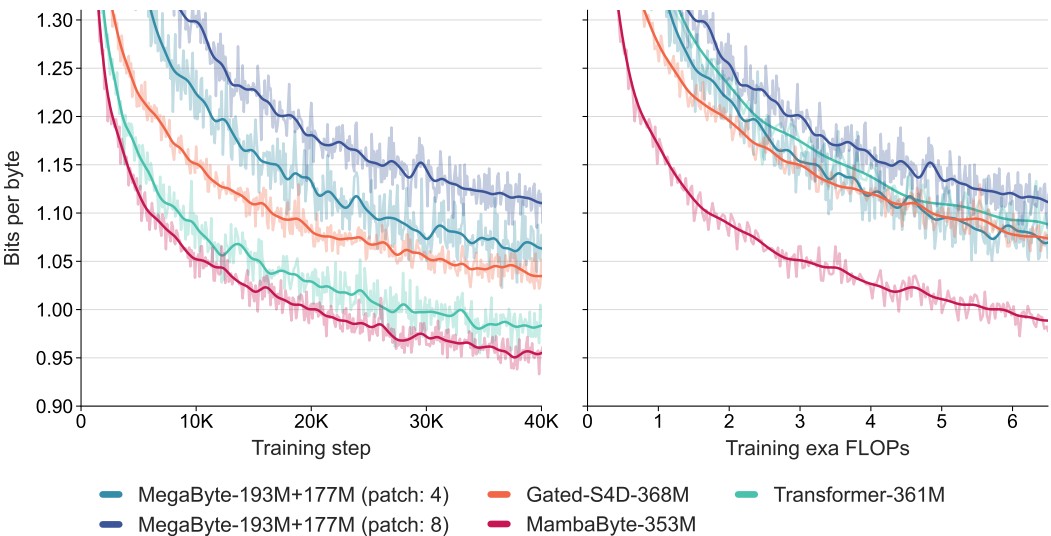

Figure 1: **Benchmarking byte-level models with a fixed parameter budget.** Language modeling results on PG19 (8, 192 consecutive bytes), comparing the standard Transformer (Vaswani et al., 2017; Su et al., 2021), MegaByte Transformer (Yu et al., 2023), gated diagonalized S4 (Mehta et al., 2023), and MambaByte. (Left) Model loss over training step. (Right) FLOP-normalized training cost. MambaByte reaches Transformer loss in less than one-third of the compute budget.

in combination with a byte-level decoder. These methods lower computational costs[1] but change the modeling behavior to match the data.

In this work, we propose MambaByte, a byte-level language model without representational compression. The model is an application of the recently introduced Mamba architecture (Gu & Dao, 2023). Mamba builds off the approach pioneered by state space models (SSMs) (Gu et al., 2021; Gupta et al., 2022; Gu et al., 2022; Smith et al., 2023; Fu et al., 2022) by introducing a selection mechanism that has been shown to be nearly as effective as Transformers for discrete data. Our key observation is that, unlike Transformers, Mamba has a (large) fixed-sized memory state that is independent of context length, roughly analogous to a large recurrent neural network hidden state. This naturally removes a major modeling and efficiency issue for byte-level language modeling without requiring specialized architectures such as global patching.

Even with effective training, byte-level models still suffer from the challenge of efficient decoding, as generating one character at a time requires running the language model in serial one byte at a time. To improve the inference efficiency, we propose an adaptation of speculative decoding (Leviathan et al., 2023; Chen et al., 2023a; Xia et al., 2023) to byte-level models. The approach uses a fast subword model for autoregressive drafting, followed by byte-level verification. While this approach could be applied to any byte-level model, it is particularly efficient for SSM-style models since the byte-level verification step can use the same parallel scan code path that makes these models efficient to train.

Experiments compare MambaByte to Transformers, SSMs, and MegaByte (patching) architectures in a fixed parameter and fixed compute setting on several long-form language modeling datasets. Figure 1 summarizes our main findings. Compared to byte-level Transformers, MambaByte achieves better performance faster and is significantly more compute-efficient. We also compare MambaByte with tokenized subword baselines using Transformers and SSMs, and find that MambaByte is competitive in loss while also demonstrating improved robustness in handling subword noise, such as input text corruptions.

---

[1]However, our experiments (see Figure 1) indicate that patching can also lower the model performance compared to the standard Transformer.

Through our speculative subword drafting and byte-level verification approach, we show that MambaByte can be run as fast as the subword Mamba for text generation. We believe these results validate the potential for tokenizer-free models as a practical alternative to subword Transformers for language modeling.

## 2 State space models and the Mamba architecture

**Method: Selective SSMs.** SSMs model the evolution of a hidden state across time through a first-order differential equation. Linear time-invariant SSMs (Gu et al., 2021; Gupta et al., 2022; Gu et al., 2022; Smith et al., 2023) have shown promising results in deep learning across several modalities. However, Gu & Dao (2023) have recently argued that the constant dynamics of these approaches lack input-dependent context *selection* in the hidden state, which may be necessary for tasks such as language modeling. To this end, they define the time-varying continuous state dynamics for a given input $x(t) \in \mathbb{R}$, hidden state $h(t) \in \mathbb{R}^n$, and output $y(t) \in \mathbb{R}$ at time $t$ as:

$$\frac{\mathrm{d}h(t)}{\mathrm{d}t} = \mathrm{A}h(t) + \mathrm{B}(t)x(t); \quad y(t) = \mathrm{C}(t)h(t), \tag{1}$$

which is parameterized by a diagonal time-invariant system matrix $\mathrm{A} \in \mathbb{R}^{n \times n}$ and time-dependent input and output matrices $\mathrm{B}(t) \in \mathbb{R}^{n \times 1}$ and $\mathrm{C}(t) \in \mathbb{R}^{1 \times n}$.

To model discrete-time sequences, the continuous-time dynamics in (1) must be approximated through discretization. This results in a discrete-time hidden state recurrence with new matrices at each timestep, $\overline{\mathrm{A}}$, $\overline{\mathrm{B}}$, and $\overline{\mathrm{C}}$, such that

$$h[k] = \overline{\mathrm{A}}[k]h[k-1] + \overline{\mathrm{B}}[k]x[k]; \quad y[k] = \overline{\mathrm{C}}[k]h[k]. \tag{2}$$

Observe that (2) resembles a linear version of a recurrent neural network and can be applied in this recurrent form during language model generation. The discretization requires a timestep, $\Delta[k]$, for each input position, corresponding to treating $x[k] = x(t_k)$ for $t_k = \sum_{j=1}^{k} \Delta[j]$. The discrete-time matrices $\overline{\mathrm{A}}$, $\overline{\mathrm{B}}$, and $\overline{\mathrm{C}}$ can then be computed from $\Delta[k]$.

**Architecture: Mamba.** In Mamba, the SSM terms are input-selective, i.e., B, C, and $\Delta$ are defined as functions of the input $x[k] \in \mathbb{R}^d$:

$$\Delta[k] = \mathrm{softplus}(W_\Delta(W_R x[k])); \quad \mathrm{B}(t_k) = W_\mathrm{B} x[k], \tag{3}$$

where $W_\mathrm{B} \in \mathbb{R}^{n \times d}$ (C is similarly defined), $W_\Delta \in \mathbb{R}^{d \times r}$ and $W_R \in \mathbb{R}^{r \times d}$ (for some $r \ll d$) are learnable weights, and softplus ensures positivity. Note that the SSM parameters A, B, and C are identical for each input dimension $d$, but the timesteps $\Delta$ are distinct; this results in a hidden state size of $n \times d$ per timestep $k$. (See Appendix D for an illustration of how Mamba models discrete sequences and other specifics on discretization and selectivity.)

Mamba embeds this SSM layer into a full neural network language model. Specifically, the model utilizes a stack of gated layers inspired by the previous gated SSM (Mehta et al., 2023). Figure 5 (right) in Appendix B shows the Mamba architecture combining the SSM layer with a gated neural network.

**Implementation: Parallel scans for linear recurrences.** At training time, we have access to the entire sequence $x$, allowing us to compute the linear recurrence more efficiently. Smith et al. (2023) demonstrated the use of work-efficient parallel scans (Blelloch, 1990) for efficiently computing the sequential recurrence in linear SSMs. For Mamba, we first map the recurrence to a sequence of $L$ tuples, with $e_k = (A_k, b_k) := (\overline{\mathrm{A}}[k], \overline{\mathrm{B}}[k]x[k])$, then define an associative operator $\bullet$ such that $e_j \bullet e_k = (A_k A_j, A_k b_j + b_k)$. Finally, we apply a parallel scan to compute the sequence $[(\overline{\mathrm{A}}[1], h[1]), (\overline{\mathrm{A}}[2]\overline{\mathrm{A}}[1], h[2]), \ldots]$. In general, this requires $\mathcal{O}(T_\bullet \log_2(L))$ time, using $L/2$ processors, where $T_\bullet$ is the cost of a matrix-matrix multiplication. Noting $\overline{\mathrm{A}}$ to be a diagonal matrix, the linear recurrence can be computed parallelly in $\mathcal{O}(n \log_2(L))$ time and $\mathcal{O}(nL)$ space. A parallel scan with a diagonal matrix is also efficient in operation, requiring $\mathcal{O}(nL)$ FLOPs.

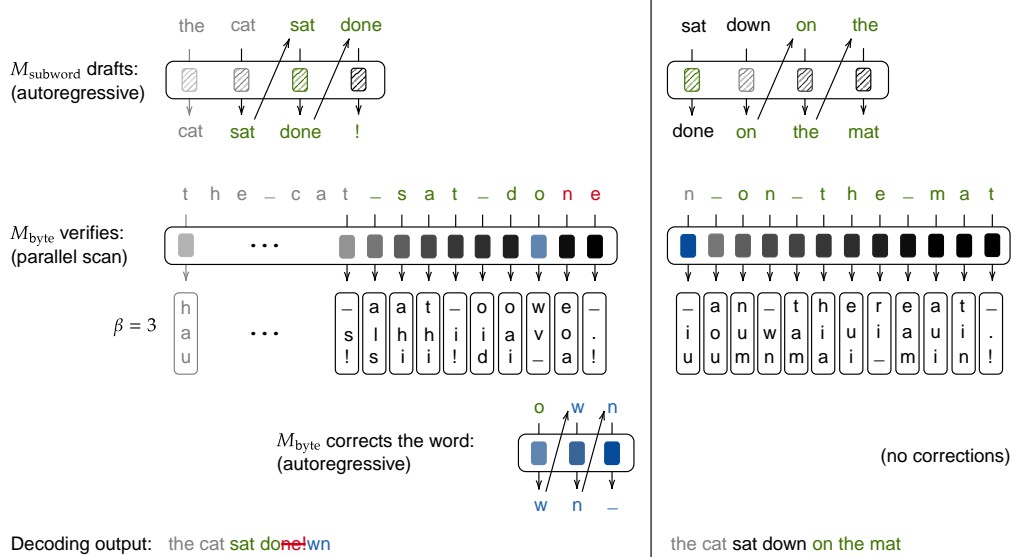

Figure 2: **Speculative decoding through subword drafting and byte-level verification.** The green subwords are suggestions made by the smaller subword (Mamba) model $M_{\text{subword}}$, whose associated bytes fell in the top-$\beta$ autoregressive candidates of the byte-level verifier (MambaByte) model $M_{\text{byte}}$. The red and blue bytes are the rejections and corresponding corrections made by the verifier model. (Two steps shown using the prompt: "the cat".)

## 3   Method

**Modeling long byte-sequences.**   MambaByte is an application of the Mamba architecture to byte-level language modeling. Our main observation is that unlike Transformers, whose memory scales linearly in sequence length, Mamba maintains a large fixed-size memory state, which makes it suitable for direct byte-level modeling. Formally, an $m$-layer Mamba model, each with a hidden state $h(t) \in \mathbb{R}^{n_{\text{state}} \times d}$, efficiently maintains and evolves a memory of $m \times n_{\text{state}} \times d$ floats. Noting that the Mamba hidden state memory size is independent of input context length, $L_{\text{ctx}}$, processing subword sequences or byte sequences requires the underlying model to compress roughly $L_{\text{ctx}}$ bytes in its fixed hidden state memory, irrespective of the input representation. In all but extreme cases, $m \times n_{\text{state}} \times d \gg L_{\text{ctx}}$, leaving enough space of a hidden state $h(t)$ to encode $L_{\text{ctx}}$ information. Therefore, if Mamba can be used for tokenized models, MambaByte should enable modeling byte-level sequences without the need for length-compression trade-offs (Dai et al., 2020; Nawrot et al., 2022; Yu et al., 2023).

Utilizing a fixed-sized memory representation may also help avoid quadratic dependencies and improve generalization. While Transformers are designed to capture long-range dependencies, researchers have noted that the sheer number of potential interactions in a long byte-level sequence can dilute the model's focus, making it challenging to capture crucial dependencies amid a vast number of less relevant ones (Tworkowski et al., 2024). Bytes level information is much more granular, thus necessitating the model to learn from a much larger context to make meaningful predictions.

Finally, training Mamba for long byte-sequences has an inherent computation benefit at scale. The computational cost for Mamba at training is $\mathcal{O}(L_{\text{ctx}})$, while even compressed models such as MegaByte (Yu et al., 2023) have a complexity of $\mathcal{O}(L_{\text{ctx}}^2/p^2 + L_{\text{ctx}}p)$ for a patch size $p$. Even with a large patch size of $L_{\text{ctx}}^{1/3}$, the resulting complexity is $\mathcal{O}(L_{\text{ctx}}^{4/3})$.

**Speculative decoding through subword drafting.**   While MambaByte is computationally efficient at training, it encounters challenges in decoding, primarily because each byte is processed sequentially. To mitigate this sequential bottleneck, we propose a adaptation of

speculative decoding through subword drafting and byte-level verification. Our observation is that most inference steps do not require the granularity of byte-level decoding and can benefit from faster subword drafting. Consequently, we can train token-free models, which are known to be robust to noise, and simulate subword-model-like generation, which is significantly faster. We decompose every decoding iteration into two steps: *draft* using a smaller subword (Mamba) model, then *verify and correct* using a larger byte-level (MambaByte) model, as illustrated in Figure 2.

The subword Mamba model, $M_{\text{subword}}$, drafts $m$ subwords autoregressively while recording the associated hidden states at each timestep. The drafted subwords are converted to bytes, fed to the byte-level MambaByte model, $M_{\text{byte}}$, and verified using a parallel scan. We then find the bifurcation byte position $c$, the furthest position in the byte sequence verified to be in the top-$\beta$ autoregressive candidates of $M_{\text{byte}}$. We also find the subword bifurcation position associated with $c$, i.e., the largest position of the subword whose bytes are all verified to be correct. Drafted bytes after position $c$ are discarded. Noting that the drafter is tokenized, while the verifier is token-free, we cannot just correct for $b_{c+1}$, i.e., one byte after the bifurcation position, and continue drafting—this causes issues with drafting, especially if the tokenizer cannot find the newly updated partial subword in its pre-trained vocabulary. To avoid the possibility of the corrected partial subword being marked as out-of-vocabulary, we use the verifier model to generate bytes autoregressively until a boundary byte (e.g., space) is generated. The final decoded tokens include the verified drafted subwords and the corrected subword generated by the byte-level model. We cache the final hidden state from the MambaByte verifier and the bifurcation hidden state from the subword Mamba model for the next iteration. For completeness, we provide the algorithm for speculative decoding through subword drafting in Appendix F.

To enable resuming during the parallel scan, we extended the fast CUDA kernel from Mamba (Gu & Dao, 2023), allowing verification to restart from the mismatched position instead of beginning from the start.

## 4   Experimental setup

Our experiments compare MambaByte to a range of other tokenizer-based and token-free Transformers and SSMs. All our models employ the same training recipes. We utilize a set of diverse long-form text datasets: PG19 (Rae et al., 2020), Stories (Trinh & Le, 2018), Books (Gao et al., 2020a), ArXiv (Gao et al., 2020a), and Code (Gao et al., 2020a). We consider models of different sizes: for MambaByte, this is indicated by the number of parameters in the model; for MegaByte, which is the primary baseline used, size is indicated by the number of parameters in the patched model and the unpatched generation head. Dataset sizes and average document lengths are included in Appendix A; model details are given in Appendix B.

| Expt | Models | FLOPs per train byte |
|---|---|---|
| Medium-scale | MegaByte-758M+262M : MambaByte-353M | 1.02 : 1 |
| Large-scale | MegaByte-1.3B+350M : MambaByte-972M | 0.54 : 1 |
| | MegaByte-1.3B+218M : MambaByte-972M | 0.40 : 1 |

Table 1: **Relative training FLOPs by model size.** MegaByte models use a patch size of 8.

Performance comparison across architectures requires care. To this end, we consider two settings: compute-matched and parameter-matched. This setup is necessary as the default MegaByte Transformer employs a global module that works with $8\times$-patched representations of the input, thus using $8\times$ fewer feed-forward FLOPs per byte than a raw Transformer, while having significantly more parameters. Table 1 shows the MegaByte and MambaByte model sizes employed in our experiments. The (forward pass) FLOPs computation for various model architectures and the associated hyperparameters employed are detailed in Appendix B.

| Byte-level model | Context | Bytes trained | Test BPB ↓ | | | | |
|---|---|---|---|---|---|---|---|
| | | | PG19 | Stories | Books | ArXiv | Code |
| Transformer-320M | 1,024 | 80B | 1.057 | 1.064 | 1.097 | 0.816 | 0.575 |
| PerceiverAR-248M | 8,192 | 80B | 1.104 | 1.070 | 1.104 | 0.791 | 0.546 |
| MegaByte-758M+262M (patch: 8) | 8,192 | 80B | 1.000 | 0.978 | 1.007 | 0.678 | 0.411 |
| MambaByte-353M | 8,192 | 30B* | **0.930** | **0.908** | **0.966** | **0.663** | **0.396** |

Table 2: **Medium-scale token-free experiments.** MegaByte-758M+262M and MambaByte-353M use the same FLOPs per byte. (The BPB for Transformer, PerceiverAR, and MegaByte are from Yu et al. (2023).)

All MambaByte models were trained using the open-source Mamba code base.[2] At training, we shuffle the documents and use contiguous sequences of 8,192 bytes (one per document), starting from a random position. We enable mixed precision training using BF16 for training efficiency at scale. The optimizer, learning rate scheduler, and other training details are specified in Appendix C.

## 5 Results

### 5.1 Language modeling performance

Table 2 shows language modeling performance in bits per byte (BPB) across each dataset. For this experiment, the MegaByte-758M+262M and MambaByte models use the same number of FLOPs per byte (see Table 1). We observe MambaByte to outperform MegaByte consistently across all datasets. Furthermore, MambaByte outperforms MegaByte with 0.63× less compute and training data. Additionally, MambaByte-353M also outperforms byte-level Transformer and PerceiverAR.

Figure 1 further explores this relationship by looking at models with the same number of parameters. The graphs indicate that for MegaByte models of the same parameter size, models with less input patching perform better, but when compute-normalized, they perform similarly. In fact, a full-length Transformer, while slow in an absolute sense, also performs similarly to the MegaByte model when compute-normalized. In contrast, switching to the Mamba architecture significantly improves both the compute usage and the model performance.

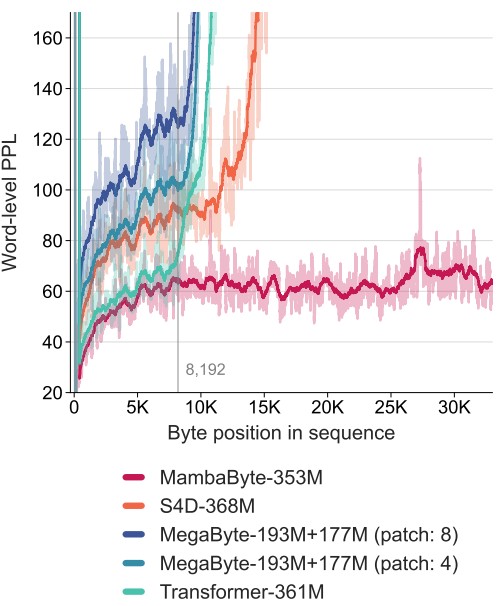

Figure 3: **Length extrapolation.** All models are trained with 8,192-long byte sequences. MambaByte can extrapolate to much longer sequences without performance degradation.

Following these findings, Table 3 compares a larger version of these models on the PG19 dataset, both with and without tokenization. For this experiment, we compare MambaByte-972M with MegaByte-1.3B+350M and other byte-level models, as well as several state-of-the-art subword models. (The conversion from BPB and subword-level perplexity to word-level perplexity (PPL) is detailed in Appendix E).

---

[2] https://github.com/state-spaces/mamba.

|  | (#Layers) Model | Vocab | Effective ctx (in bytes) | Effective bytes trained | Val PPL↓ | Test PPL↓ |
|---|---|---|---|---|---|---|
| Subword | (36) Transformer-XL (Rae et al., 2020) | 32K | 2,048/4,096 | 400B | 45.5 | 36.3 |
|  | (36) Compressive (Rae et al., 2020) | 32K | 2,048/2×2,048 | 400B | 43.4 | 33.6 |
|  | (22) Routing-490M (Roy et al., 2021) | 82K | 32,768 | 330B | − | 33.2 |
|  | (60) PerceiverAR-974.6M (Hawthorne et al., 2022) | 32K | 8,192 | 1.68T | 45.9 | 28.9 |
|  | (24) Block-Recurrent-1.3B (Hutchins et al., 2022) | 32K | 4,096/recurrence | − | − | **26.5** |
|  | (48) Mamba-1.03B | 32K | 8,192 | 600B* | 40.1 | 33.9 |
| Byte | (−) Transformer-320M (Yu et al., 2023) | 256 | 8,192 | 400B | 81.6 | 69.4 |
|  | (−) PerceiverAR-248M (Yu et al., 2023) | 256 | 8,192 | 400B | 119.1 | 88.8 |
|  | (24+24) MegaByte-1.3B+350M (Yu et al., 2023) | 256 | 8,192/patch: 8 | 400B | 42.8 | 36.4 |
|  | (48) MambaByte-972M | 256 | 8,192 | 150B* | **39.6** | 33.0 |

Table 3: **Large-scale experiment on PG19.** The observed BPB scores are converted to word-level PPL for comparison with past works. All the byte-level models are compute-matched. Mamba-1.03B and MambaByte-972M are evaluated using 4× longer context and a sliding window of 16,384 bytes. MambaByte-972M significantly outperforms other byte-level models and is competitive with state-of-the-art subword models. (Accompanying citation indicates the work from which the corresponding result was taken; fields marked − are unknown.)

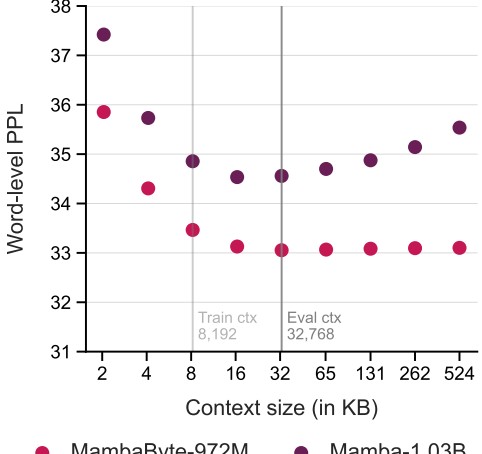

Figure 4: **Long context experiment.** Length extrapolation using a sliding window of $L_{ctx}/2$.

| | Proba | PPL ↓ | |
|---|---|---|---|
| | | Mamba | MambaByte |
| Drop | 0.05 | +16.9 | **+8.5** |
| | 0.3 | +213.2 | **+31.7** |
| Repeat | 0.05 | +6.3 | **+6.2** |
| | 0.3 | +28.4 | **+26.6** |
| Antspeak | | +58300.0 | **+28.3** |
| Uppercase | 0.05 | +5.4 | **+1.6** |
| | 0.3 | +18.3 | **+5.5** |
| Random case | | +20.8 | **+7.7** |
| Swap | 0.05 | +29.0 | **+9.3** |
| | 0.3 | +630.6 | **+28.7** |

Table 4: **Noise experiments.** Degradation of Mamba-1.03B and MambaByte-972M under varied noise settings.

We find that MambaByte-972M, even just trained for 150B bytes, outperforms all the byte-level models and achieves competitive performance with subword models.

Figure 3 records another interesting aspect of MambaByte: its ability to extrapolate to significantly longer sequences (at least 4× longer than the training length) compared to other byte-level Transformer and SSM baselines, suggesting that MambaByte can effectively refine the recurrent hidden state for significantly longer sequences. As expected, limited by the position embeddings, Transformer models don't extrapolate beyond the training length.

## 5.2 Token-free capabilities

To control for the benefits of the Mamba architecture, we retrained a subword Mamba-1.03B model in a compute-matched setting (see Table 3). Interestingly, the (subword) Mamba and MambaByte perform similarly at the same parameter size and training compute. As previously mentioned, these models effectively have the same memory capacity despite

| Model | Bytes trained | Context | Test BPB ↓ | Generation time (s) ↓ |
|---|---|---|---|---|
| Transformer-350M | 80B | $1,024$ | 1.064 | 132 |
| MegaByte-1.3B+218M | 80B | $8,192$ | 0.991 | 93 |
| MegaByte-1.3B+218M | — | $8,192$ | — | 265 |
| MambaByte-972M | 75B* | $8,192$ | **0.883** | **29** |
| w/ sliding window (2× bytes) | | | **0.863** | 58 |
| MambaByte-1.6B | — | $8,192$ | — | 36 |

Table 5: **Generation speed benchmarking.** Speed to generate $8,192$ bytes; fields marked − are unknown. (Upper) The BPB on PG19 and generation time for the Transformer and MegaByte are taken from Yu et al. (2023). (Lower) MegaByte and MambaByte run on the same hardware; we use the available open-source MegaByte implementation here.

significant differences in the input sequence length. We also find that Mamba achieves near-optimal performance more efficiently than MambaByte, though not 4× faster as expected, but 2.2× faster. Furthermore, the perplexity for Mamba-1.03B does not improve significantly beyond 150B training bytes, consistent with the observations made by Rae et al. (2020). Given the similar performance of Mamba and MambaByte, we can further explore downstream capabilities.

**Modeling longer contexts.** From Figure 4, we note that Mamba and MambaByte show impressive extrapolation capabilities for sequences up to 64× longer than the training length. We hypothesize that MambaByte shows slightly better length extrapolation than the subword Mamba because MambaByte models 4× longer sequences at training despite both models processing the same effective number of bytes per training sequence.

**Synthetic noise experiments.** We employ the synthetic noise benchmark from Xue et al. (2022) to test model robustness; additional details about the noise settings are noted in Appendix G. We process the input text in the PG19 test set into chunks of 100 space-separated words and inject noise into every odd-indexed chunk while retaining the text in the even-indexed chunk unaltered. Table 4 shows the degradation of word-level PPL with noise injection, measured on even-indexed chunks. We observe that Mamba performance degrades significantly in the presence of noise compared to MambaByte across all noise conditions, indicating that tokenized vocabulary fundamentally limits subword models. This effect is pronounced in specific noise settings such as antspeak (every character is capitalized and padded with spaces) and character swapping (consecutive bytes are swapped). Our findings align with those observed by Xue et al. (2022) in that byte-level models are significantly more resilient to accidental and adversarial spelling errors than subword models.

### 5.3 Generation efficiency

Autoregressive inference in Transformer models requires caching the entire context, which can significantly affect the generation speed. MambaByte does not suffer from this bottleneck as it maintains a single hidden state per layer that evolves with time, enabling constant time per generation step. Table 5 compares the text generation speeds of MambaByte-972M and MambaByte-1.6B with MegaByte-1.3B+350M on an A100 80GB PCIe GPU. While MegaByte significantly reduces the generation cost through patching, we observe MambaByte to be 2.6× faster in a parameter-matched setting due to its use of recurrent generation. Appendix H includes more information about the generation process.

**Generation via subword speculation.** Table 6 shows the inference speedup using speculative decoding through subword drafting, averaged across 100 prompts generated using common phrases in the PG19 dataset. We use a Mamba-110M model as the drafter, and the subwords are generated using greedy decoding. We observe that through speculative

| Model | Context | Relative speedup ↑ | Log-odds ratio ↑ |
|---|---|---|---|
| MambaByte-972M | 8,192 | 1× | 1.0 |
| Mamba-1.03B | 2,048 | 2.8× | 0.10 |
| MambaByte-972M w/ Mamba-110M speculation | 8,192 | **2.6×** | **0.89** |

Table 6: **Generation speed with subword speculation.** Empirical results for speeding up inference from the MambaByte-972M model. Speed was measured in generating 8,192 bytes; the drafter drafts three subwords per iteration and the verifier accepts if the drafted bytes were in its top-3 candidates.

subword drafting, MambaByte can achieve a decoding speed nearing that of the subword Mamba. Furthermore, to assess the faithfulness of our speculative decoding approach, we use a greedy-decoded MambaByte-972M generation as the reference candidate for a given prompt. We report the ratio of the log-likelihood of generating the reference candidate to the log-likelihood of MambaByte generating the speculative-decoded sequence, which is averaged across all prompts. From Table 6, we observe our speculative decoding approach to be more faithful to MambaByte-972M than the subword Mamba-1.03B.

## 6 Related work

**Token-free language models.** Tokenization has been fundamental to language modeling and vital in enhancing model efficiency and understanding. Several algorithms have been proposed to address tokenization issues, including sizeable vocabulary size and handling out-of-vocabulary tokens: Byte-Pair Encoding (Sennrich et al., 2015), WordPiece (Schuster & Nakajima, 2012; Devlin et al., 2018), and SentencePiece (Kudo & Richardson, 2018). The recent shift towards character (Tay et al., 2022; Ma et al., 2020; Mielke & Eisner, 2019) and byte-level (Yu et al., 2023; Xue et al., 2022; Belouadi & Eger, 2022) modeling aims to achieve token-free preprocessing, thereby facilitating improved model adaptability and domain transferability in language modeling and multilingual processing.

**Attention-free models.** Attention-free models offer enhanced computational and memory efficiency and are increasingly adapted for several language processing tasks, including autoregressive language modeling. Models such as S4 (Gu et al., 2021) and its subsequent variants (Gupta et al., 2022; Gu et al., 2022) have demonstrated promising outcomes in subword-level language modeling. Gated SSM architectures such as GSS (Mehta et al., 2023) and BiGS Wang et al. (2022) incorporated a gating mechanism into SSMs for (bidirectional) language modeling. The recently introduced Mamba model (Gu & Dao, 2023) posits that the unchanging dynamics of these methods fail to incorporate input-specific context selection within the hidden state, which might be crucial for tasks like language modeling. Mamba has been shown to outperform Transformers across model sizes and at scale. Alternatively, several other sub-quadratic model architectures (Yang et al., 2023b; De et al., 2024; Arora et al., 2023; 2024; Fu et al., 2024a) have also been proposed. Beyond language modeling, SSMs and Mamba have been applied in other modalities, including images (Yan et al., 2024), audio (Goel et al., 2022), and bioinformatics (Schiff et al., 2024).

**Speculative decoding for fast inference.** Speculative decoding (Spector & Re, 2023; Leviathan et al., 2023; Chen et al., 2023a; Xia et al., 2023) has emerged as a promising approach to accelerate the inference of large language models, specifically Transformers. The core idea is to use a smaller draft model to speculatively generate candidate tokens, which the larger target model then verifies. Leviathan et al. (2023); Chen et al. (2023a) proposed a rejection sampling scheme to improve the inference quality. Spector & Re (2023) restructured the candidate tokens into a tree to enable more efficient verification. Additional approaches also investigated trained draft models (Bhendawade et al., 2024; Chen et al., 2023b; Liu et al., 2023) and training-free draft models (He et al., 2023; Yang et al., 2023a; Fu

et al., 2024b). While previous methods employ drafter and verifier models with the same underlying tokenization scheme, this paper proposes using a smaller subword Mamba model as the speculative drafter and a larger MambaByte as the byte-level verifier.

## 7 Conclusion

We introduce MambaByte, a token-free SSM for modeling long byte-sequences. MambaByte outperforms other byte-level models over several datasets and shows competitive results with subword Transformers while being significantly robust to text corruptions, thus serving as a promising tokenization alternative. Due to their recurrent nature, SSMs enable significantly faster text generation to Transformer models. We further improve the generation efficiency through speculative decoding using subword drafting and show MambaByte to achieve to achieve a decoding efficiency similar to the subword Mamba, making byte models practical. Our findings establish the possibility of token-free language modeling in future large models.

## Acknowledgments

We thank Albert Gu for their helpful comments on MambaByte, Tri Dao for their guidelines on extending the selective scan kernel in Mamba, and the authors of MegaByte, Lili Yu and Mike Lewis, for clarifications on MegaByte training and inference procedures. This work was supported by NSF IIS-1901030 and NSF CAREER 2037519.

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

## Appendix

## A  Dataset specifics

|       | Total bytes | Total docs | Bytes/doc |
|-------|-------------|------------|-----------|
| PG19    | 11.74G   | 28,752      | 4,082,210 |
| Stories | 34.18G   | 948,247     | 36,045    |
| Books   | 108.38G  | 196,640     | 551,179   |
| ArXiv   | 60.27G   | 1,264,405   | 47,665    |
| Code    | 677G     | 56,626,342  | 11,958    |

Table 7: **Text dataset statistics.** The total bytes, total documents, and the mean document size (bytes per document) for each dataset.

We benchmark our results on various long-form text datasets. The PG19 dataset (Rae et al., 2020) is an extensive collection of full-length English books (written before 1919) from the Project Gutenberg online library. The PG19 dataset is ideal to test for long-distance context modeling (Gao et al., 2020a). The Stories dataset (Trinh & Le, 2018) is a subset of the CommonCrawl data used for commonsense reasoning and language modeling. The Books dataset (Gao et al., 2020a) is another collection of English books. The ArXiv dataset (Gao et al., 2020a) comprises technical publications in LATEX from the arXiv online archive. Finally, the Code dataset (Gao et al., 2020a) is a large dataset of publicly available open-source code (under Apache, MIT, or BSD licenses). Dataset statistics are tabulated in Table 7.

For the PG19 dataset, we employ the train, validation, and test data splits as indicated by Rae et al. (2020). For Stories, Books, ArXiv, and Code datasets, we randomly sample 40M consecutive bytes for testing and the rest to train MambaByte.

## B  Compute-constrained modeling

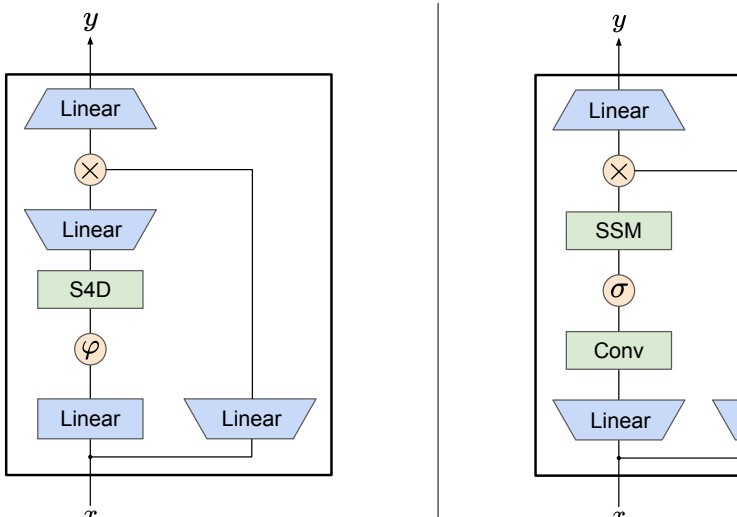

Figure 5: **SSM model network architectures.** (Left) Gated-S4D block adapted from Mehta et al. (2023); (right) Mamba SSM block. ($\varphi$ indicates GELU activation (Hendrycks & Gimpel, 2016), and $\sigma$ indicates Swish activation (Ramachandran et al., 2017).)

As noted earlier, we evaluate and benchmark MambaByte in a compute-controlled setting. To this end, we estimate the FLOPs per byte incurred by various byte-level model architectures. We parameterize the architectures using hyperparameters $n$ ($n_g/n_l$) number of (global/local) layers, dimension $d$ ($d_g/d_l$) of the (global/local) residual stream, expansion factor $e$ of linear layers, patch size $p$ in MegaByte, state dimension $n_{\text{state}}$ in SSMs, 1D convolution kernel size $k$, and low-rank projection dimension $r$ in Mamba. We also include $L_{\text{ctx}}$ bytes in the input context. Detailed component-wise compute counts for the forward pass are included in Table 8.

| Model | Component | FLOPs per byte |
|---|---|---|
| Transformer (Vaswani et al., 2017) | Multi-head attention | $2n(4d^2 + 2L_{ctx}d)$ |
| | Pointwise feed-forward | $2n(2ed^2)$ |
| MegaByte[3] (Yu et al., 2023) | Embedding projection | $2d_g^2$ |
| | Global transformer model | $2n_g(4d_g^2 + 2d_g L_{ctx}/p + 2ed_g^2)/p$ |
| | Global-to-local projection | $2d_g d_l$ |
| | Local transformer model | $2n_l(4d_l^2 + 2pd_l + 2ed_l^2)$ |
| Gated-S4D (Figure 5, left) | Linear projections | $2n(3ed^2 + d^2)$ |
| | Kernel via Vandermonde $v(\overline{A})$ | $n(\alpha_v ed(n_{state} + L_{ctx})\log_2^2(n_{state} + L_{ctx})/L_{ctx})$ |
| | S4D SSM with convolution | $n(\alpha_{fft}\log(L_{ctx})ed + ed)$ |
| | Element-wise gating | $ned$ |
| MambaByte (Figure 5, right) | Linear projections | $2n(3ed^2)$ |
| | Pre-SSM 1D convolution | $2nked$ |
| | $\Delta, B, C$ from input $x$ | $2n(2edr + 2edn_{state})$ |
| | Discretization, pre-scan: $\overline{A}, \overline{B}x$ | $n(3edn_{state})$ |
| | Recurrence with parallel scan | $n(edn_{state})$ |
| | Output: $y = \overline{C}h + \overline{D}x$ | $2nedn_{state} + ned$ |
| | Element-wise gating | $ned$ |

Table 8: **Compute (forward pass) estimates for various byte-level language models.** Embedding, de-embedding, and sub-leading terms such as biases, nonlinearities, and layer norms are omitted. ($\alpha_*$ indicates an implementation-specific constant scaling term.)

For the medium-scale language modeling experiments (Table 1, §5 of Yu et al. (2023)), Yu et al. (2023) employ the MegaByte-758M+262M model, with a context length of 8,192 and patch size of 8, trained for 80B bytes. As shown in Figure 6, MambaByte-353M ($n = 53$, $d = 1,024$, $e = 2$) and MegaByte-758M+262M use the same total compute in FLOPs; hence, we employ the MambaByte-353M to benchmark against MegaByte-758M+262M in Table 2 of §5.

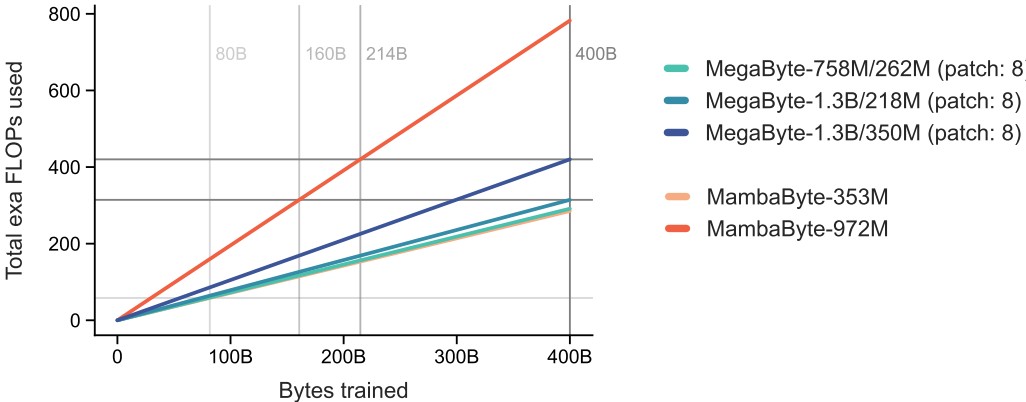

Figure 6: **Computational cost for different model architectures at different scales.** All models use a context length of 8,192, and MegaByte architectures use a patch size of 8.

For the PG19 scaling experiment (Table 2, §5 and Appendix D.3 of Yu et al. (2023)), Yu et al. (2023) use MegaByte-1.3B+350M (context length of 8,192 and patch size of 8) trained for 400B bytes to benchmark the observed word-level perplexity against several state-of-the-art subword models. Owing to our hardware limitations, we train MambaByte-972M ($n = 48$, $d = 1,792$, $e = 2$) and control for the total compute used (see Figure 6 to view the associated computational costs). All the model sizes and associated hyperparameters employed in this work are tabulated in Table 9.

| Model | Parameters | Hyperparameters | | | | |
|---|---|---|---|---|---|---|
| | | $n$ $(n_g/n_l)$ | $d$ $(d_g/d_l)$ | $e$ | $L_{\text{ctx}}$ | Others |
| Transformer | 320M (Yu et al., 2023) | 22 | 1,024 | 4 | 1,024 | heads: $-$ |
| | 350M (Yu et al., 2023) | 24 | 1,024 | 4 | 1,024 | heads: 16 |
| | 361M | 28 | 1,024 | 4 | 8,192 | heads: 16 |
| PerceiverAR 248M (Yu et al., 2023) | | 17 | 1,024 | 4 | 8,192 | latents: 1,024 |
| MegaByte | 193M+177M[3] | 14/14 | 1,024/1,024 | 4 | 8,192 | $p = 4, 8$; heads: 16/16 |
| | 758M+262M (Yu et al., 2023) | 14/18 | 2,048/1,024 | 4 | 8,192 | $p = 8$; heads: 16/16 |
| | 1.3B+218M (Yu et al., 2023) | 24/15 | 2,048/1,024 | 4 | 8,192 | $p = 8$; heads: 32/$-$ |
| | 1.3B+350M (Yu et al., 2023) | 24/24 | 2,048/1,024 | 4 | 8,192 | $p = 8$; heads: 32/16 |
| Gated-S4D | 368M | 26 | 1,024 | 4 | 8,192 | $n_{\text{state}} = 64$ |
| MambaByte | 353M | 53 | 1,024 | 2 | 8,192 | $k = 4; n_{\text{state}} = 16; r = 64$ |
| | 972M | 48 | 1,792 | 2 | 8,192 | $k = 4; n_{\text{state}} = 16; r = 112$ |
| | 1.6B | 48 | 2,304 | 2 | 8,192 | $k = 4; n_{\text{state}} = 16; r = 144$ |

Table 9: **Model hyperparameters.** We report the model size and associated hyperparameters for all the models employed in this study. (Accompanying citation indicates the work from which the associated configuration is noted; fields marked as $-$ are unknown.)

## C  Training recipes

All the models in this study were trained using an AdamW optimizer with $\beta = (0.9, 0.95)$. We used a linear learning rate warm-up (for the first 500 steps) followed by cosine annealing. Keeping consistent with MegaByte training (Yu et al., 2023), we used a batch size of 48 across all our experiments. Additionally, we do not use dropout with any of our models.

For the experiments in Figure 1, we conducted a hyperparameter search using peak learning rates of 0.0002, 0.0006, and 0.0008 and clipped the gradient norm to 1.0 for all the models. The best-observed performance curve for each model is reported in Figure 1. Furthermore, we use an improved Transformer recipe that uses RMSNorm instead of LayerNorm, rotary positional encodings (Su et al., 2021), and linear terms without bias (same as Yu et al. (2023)).

In our medium-scale experiments shown in Table 2, we set the peak learning rate to 0.0004 and clipped the gradient norm to 0.1. We trained the MambaByte-353M for a total of 80K steps, equivalent to $80,000 \times 48 \times 8,192 \approx 30\text{B}$ bytes.

In the large-scale experiment on PG19, we use a similar setting to that in the medium-scale experiments: the peak learning rate is set to 0.0004, and the gradient norm is clipped to 0.1. The MambaByte-972M is trained for 380K steps, equivalent to $380,000 \times 48 \times 8,192 \approx 150\text{B}$ bytes.

## D  Discretization and selection

Discretization has deep connections to continuous-time systems, which allows for desirable properties such as model normalization (Orvieto et al., 2023; Gu et al., 2023) and resolution invariance (Nguyen et al., 2022). In this section, we review how zero-order hold discretization of Mamba selective SSM can be viewed as a generalization of the gating mechanism in recurrent networks. An illustration of the Mamba SSM discretization and illustration is depicted in Figure 7.

**Zero-order hold discretization.**  For a given input $x(t) \in \mathbb{R}$, we wish to discretize a continuous-time SSM defined by (1) in §2. To this end, we sample the system at different time intervals such that $x[k] = x(t_k)$ for $t_k = \sum_{j=1}^{k} \Delta[j]$ and assume a zero-order hold, i.e.,

---

[3]We used the open-source implementation: https://github.com/lucidrains/MEGABYTE-pytorch.

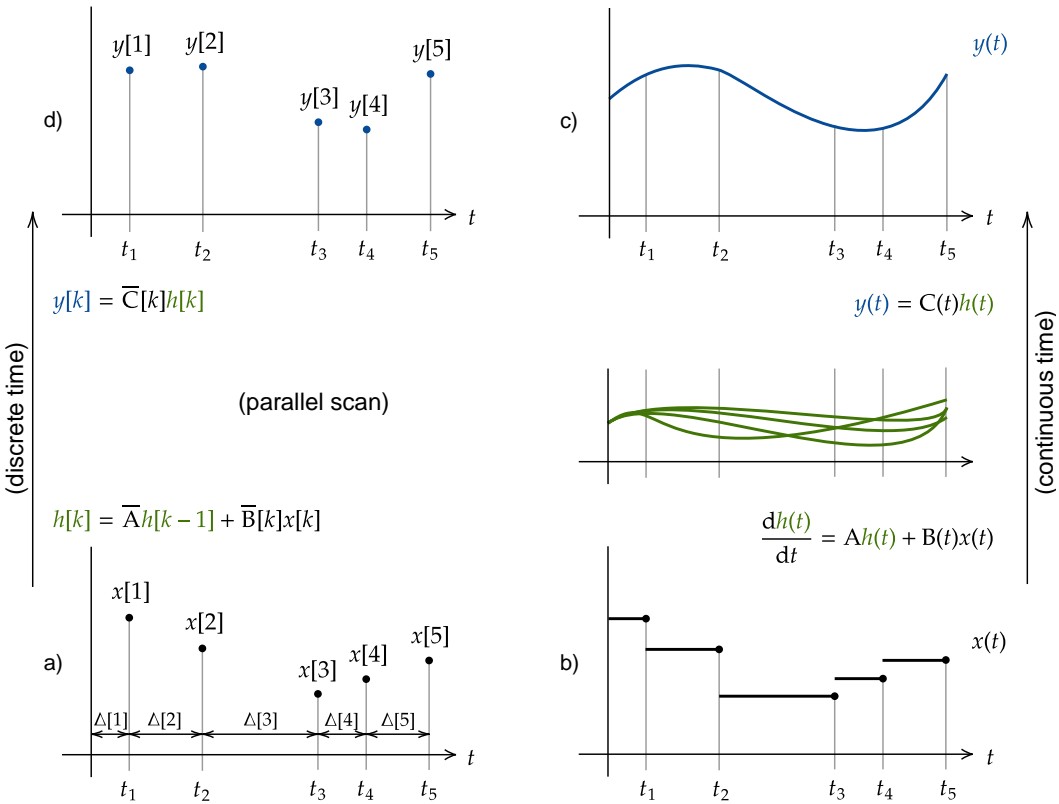

Figure 7: **Illustration of the Mamba SSM.** (a) The discrete-time input $x[k]$, along with input-selective $\Delta[k]$. (b) The continuous-time signal $x(t)$. (c) Mathematically, the SSM transforms the continuous-time $x(t)$ through an $n$-dimensional hidden state (here, $n = 4$) using parameters A and $B(t)$, which is then mapped to the output $y(t)$ using $C(t)$. (d) Practically, we compute $y[k]$ using a discrete-time parallel scan at the steps defined by $\Delta[k]$ and discrete-time matrices $\overline{A}[k]$, $\overline{B}[k]$, and $\overline{C}[k]$. At inference, we run the recurrence directly.

$x(t)$ is constant between samples: $x(t_k + \xi) = x(t_k) = x[k]$ for any $\xi \in [t_k, t_{k+1})$. The resultant matrices of the associated discrete SSM are:[4]

$$\overline{A} = \exp(A\,\Delta); \quad \overline{B} = A^{-1}(\exp(A\,\Delta) - I)\,B; \quad \overline{C} = C.$$

**Selection mechanics and gating in recurrent networks.** Gu & Dao (2023) note that a selective SSM can be realized as a gated recurrence by setting $\Delta = \text{softplus}(z(x)) = \text{softplus}(W_\Delta(W_R x))$ (as indicated in (3) of §2). By letting $A = -1$, $B = 1$, and $n = 1$, the authors observe:

$$\begin{aligned} \overline{A} &= \exp(A\,\Delta) \\ &= \exp(-\log(1 + \exp(z(x)))) \\ &= \frac{1}{1 + \exp(z(x))} \\ &= \sigma(-z(x)) \\ &= 1 - \sigma(z(x)). \end{aligned} \qquad \begin{aligned} \overline{B} &= A^{-1}(\exp(A\,\Delta) - I)\,B \\ &= I - \exp(A\,\Delta) \\ &= \sigma(z(x)). \end{aligned}$$

---

[4]In Mamba (Gu & Dao, 2023), B is discretized through a simplified Euler (as opposed to zero-order hold) discretization from empirical observations of A being more important than B, and the performance does not change significantly with simplification on B.

Using $\overline{A}$ and $\overline{B}$ from above in the discrete recurrence (2), the selective SSM takes the form of a 1D gated recurrence:

$$h[k] = (1 - \sigma(z(x)))\, h[k-1] + \sigma(z(x))x[k]. \tag{4}$$

It is interesting to note from (4) that $\lim_{\Delta \to \infty} h[k] = x[k]$ and $\lim_{\Delta \to 0} h[k] = h[k-1]$: a large $\Delta$ ($\Delta \to \infty$) denotes the evolution of the system to focus only on the current input and forgetting the state. In contrast, a small $\Delta$ ($\Delta \to 0$) represents a transient input being ignored.

**Selectivity of A, B, and C matrices.** Gu & Dao (2023) argue that since the system matrix A only affects the model through $\Delta$, i.e., $\overline{A} = \exp(A\,\Delta)$. Hence, the selectivity in $\Delta$ is sufficient to ensure selectivity in A.

While the selectivity in $\Delta$ enables selectivity in the input matrix B, Gu & Dao (2023) hypothesize that making B and C selective (in addition to $\Delta$) would allow for more fine-grained control based on the content $x[k]$ and evolving context $h[k]$.

## E    Evaluation metrics

Subword-based language models (Vaswani et al., 2017; Hawthorne et al., 2022; Hutchins et al., 2022) report their performance in word or subword-level PPL, while byte-level language models (Xue et al., 2022; Yu et al., 2023) report theirs in BPB. To facilitate meaningful comparisons, we report performance in BPB when benchmarking against byte-level models and word-level PPL when comparing to token-level models.[5] This section details the conversion from BPB and subword-level PPL to word-level PPL.

Irrespective of the underlying segmentation, the amount of information $I(D)$ in a given dataset $D$ is constant. Simply put,

$$I(D) = L_W \text{ bits per word} = L_S \text{ bits per subword} = L_B \text{ bits per byte} \tag{5a}$$

$$\triangleq \frac{-\ln(D; \text{model})}{\ln(2)}, \tag{5b}$$

where $L_W$, $L_S$, and $L_B$ are the length of the dataset in words, subwords, and bytes, respectively. From (5), we observe:

$$\text{BPB} = \frac{-\ln(D; \text{model})/L_B}{\ln(2)} = \frac{\ell_{\text{byte}}}{\ln(2)},$$

where $\ell_{\text{byte}}$ is the observed byte-level negative log-likelihood loss (computed using ln). From (5), we also note the following conversion from BPB to word-level PPL:

$$\frac{-\ln(D; \text{model})/L_W}{\ln(2)} = \frac{L_B}{L_W}\,\text{BPB} = \frac{L_B}{L_W}\frac{\ell_{\text{byte}}}{\ln(2)}$$

$$\Rightarrow \text{PPL} = \exp\left(\frac{L_B}{L_W}\ell_{\text{byte}}\right) = \exp\left(\frac{L_B}{L_W}\ln(2)\,\text{BPB}\right).$$

Similarly, we can compute word-level PPL from the observed subword-level negative log-likelihood loss $\ell_{\text{subword}}$ as:

$$\text{PPL} = \exp\left(\frac{L_S}{L_W}\ell_{\text{subword}}\right).$$

For the PG19 dataset, we train MambaByte-972M to minimize BPB over the training data and report word-level PPL on the test data. In our medium-scale benchmarking experiments, for (subword) Mamba-1.03B, we trained a 32K-subword vocabulary using the Subword-TextEncoder from the `tfds` package in TensorFlow, the same as Rae et al. (2020). Split-wise values of $L_B/L_W$ and $L_S/L_W$ for the PG19 dataset are tabulated in Table 10.

---

[5]Unless stated otherwise, we use PPL to report word-level (not subword-level) perplexity.

| | $L_B$ | $L_S$ | $L_W$ | $L_B/L_W$ | $L_S/L_W$ |
|---|---|---|---|---|---|
| Train | $11,677,824,216$ | $2,914,582,573$ | $1,973,048,393$ | $5.92$ | $1.48$ |
| Validation | $17,733,002$ | $4,357,506$ | $3,007,061$ | $5.90$ | $1.45$ |
| Test | $41,289,101$ | $10,282,006$ | $6,965,511$ | $5.93$ | $1.48$ |

Table 10: **PG19 dataset statistics.** Split-wise UTF-8 encoded byte $L_B$, SentencePiece-tokenized subword $L_S$, and space-separated word $L_W$ counts in the PG19 dataset. (The byte count includes the newline character.) We also indicate the associated bytes per word $L_B/L_W$ and subwords per word $L_S/L_W$.

# F  Speculative decoding through subword drafting algorithm

Algorithm 1 below outlines our speculative decoding approach: the smaller subword draft model drafts $m$ subwords at a time, which are then verified at a byte-level by a larger MambaByte model.

---

**Algorithm 1** Speculative decoding iteration with subword drafter and byte-level verifier and corrector. (We use $\tilde{b}$ to indicate a drafted byte, while $\hat{b}$ denotes a corrected byte.)

---

**Inputs:** $M_{\text{subword}}$, $M_{\text{byte}}$, prefix subwords $s_{1:t}$, previous $M_{\text{subword}}$ hidden state $\hbar_{\text{prev}}$, previous $M_{\text{byte}}$ hidden state $h_{\text{prev}}$, draft block size $m$, verify model tolerance $n$.

$\triangleright$ Sample $m$ draft subwords $\tilde{s}_j$s from $M_{\text{subword}}$ autoregressively and record the hidden states $\hbar_j$s at each timestep. $\triangleleft$

$\hbar_0 \leftarrow \hbar_{\text{prev}}$
**for** $j = 1, \ldots, m$ **do**
$\quad q_j(x), \hbar_j \leftarrow M_{\text{subword}}(s_{1:t} + \tilde{s}_{1:j-1}, \hbar_{j-1})$
$\quad \tilde{s}_j \sim q_j(x)$
$b_{1:t'}, \tilde{b}_{1:n} \leftarrow \text{bytes}(s_{1:t}), \text{bytes}(\tilde{s}_{1:m})$ $\quad\triangleright$ Get bytes for both prefix and drafted subwords.
$\triangleright$ Run $M_{\text{byte}}$ in parallel to verify the drafted bytes, while recording the associated hidden states $h_i$s. $\triangleleft$
$(p_1(x), h_1), \ldots, (p_n(x), h_n) \leftarrow$
$\quad M_{\text{byte}}(b_{1:t'}, h_{\text{prev}}), M_{\text{byte}}(b_{1:t'} + \tilde{b}_1), \ldots, M_{\text{byte}}(b_{1:t'} + \tilde{b}_{1:n-1})$
$\triangleright$ Find the bifurcation position $c$ such that $\tilde{b}_{1:c}$ drafted bytes all fall in top-$\beta$ candidates of $M_{\text{byte}}$, while $\tilde{b}_{c+1}$ does not. $\triangleleft$
$c \leftarrow \min\left(\{i \mid 1 \leq i \leq n, \text{rank}_{p_i}(\tilde{b}_i) > \beta\} \cup \{n\}\right)$
$c' \leftarrow \min\left(\{j \mid 1 \leq j \leq m, \text{cumsum}(\text{len}(\tilde{s}_{1:m}))[j] > c\} \cup \{m\}\right)$ $\triangleright$ Find associated subword bifurcation position.
$\triangleright$ Starting from $\tilde{b}_c$ (and using $h_c$), sample corrected bytes $\hat{b}_i$s from $M_{\text{byte}}$ autoregressively until a boundary byte (e.g., space) is obtained. $\triangleleft$
$\hat{b}_c \leftarrow \tilde{b}_c; k \leftarrow 0$
**while** $\hat{b}_{c+k}$ is not a boundary byte **do**
$\quad k \leftarrow k+1$
$\quad p_{c+k}(x), h_{c+k} \leftarrow M_{\text{byte}}(\hat{b}_{c+k-1}, h_{c+k-1})$
$\quad \hat{b}_{c+k} \sim p(x)$
**return** generated bytes $\tilde{b}_{1:c} + \hat{b}_{c+1:c+k}$, $M_{\text{byte}}$ last hidden state $h_{c+k}$, $M_{\text{subword}}$ hidden state $\hbar_{c'}$ as cache to restart from.

---

# G  Synthetic noise settings

To confirm the robustness of MambaByte to input text corruptions, we employ the following synthetic noise settings adapted from Xue et al. (2022):

1) *Drop*: Bytes are dropped with a pre-set probability.

2) *Repetition*: Bytes are repeated one to three times (with equal likelihood).

3) *Antspeak*: Every character is capitalized and padded with spaces.

4) *Uppercase*: Characters are converted to uppercase with a pre-set probability.

5) *Random case*: Every character is set to a random case.

In addition to these five settings, we include the *character swap* setting, where consecutive bytes are swapped with some probability.

## H   PG19 generation samples

This section includes a few sample generations from the MambaByte-972M trained on the PG19 dataset. We use Nucleus sampling with $p = 0.98$ (Holtzman et al., 2020) and generate continuations for a total of $8,192$ bytes (including the given context prefix). Furthermore, we chose the same test set prefixes used in Appendix F of Rae et al. (2020). We observe that the model is able to continue the dialogue in the style of the prefix and effectively recall the character names over hundreds of bytes.

**Baby Mine by Margaret Mayo**

Context (487 bytes):

> CHAPTER I
>
> Even in college Alfred Hardy was a young man of fixed ideas and high ideals and proud of it.
>
> His friend, Jimmy Jinks, had few ideas and no ideals, and was glad of it, and before half of their first college term had passed, Jimmy had ridded himself of all such worries as making up his own mind or directing his own morals. Alfred did all these things so much better, argued Jimmy, furthermore, Alfred LIKED to do them–Jimmy owed it to his friend to give him that pleasure.

MambaByte-972M:

> It was not that Alfred really considered his friend an ideal husband, but Jimmy admitted that it looked differently. And, too, it was all there was to life.
>
> And on went Alfred, through long years of honest comradeship and unblushing promises and bubbling kindness. In his heart he had long since forgotten Jimmy and his friendship, since he realized the superiority of Jimmy's ideal, if of any one's.
>
> It was to be expected that Prince William would fall a victim. Perhaps he was a modified betting man, and then had thrown over Nature and his team-mates, as a victim to ridicule or a follower of hazardous ways. It was then that Prince William went back into society, perhaps sought an interview with Miss La Roche, and learned that he had been the victim of romance death, and a suicide, and had been endowed with an absurd roman cross. Whatever it was, he saw in it the myth of Alfred's heroic deeds or the myth of a yellow dog in the market plate. It sobered him gradually. He became nearly a convert, and it was Alfred who, in bidding Miss Rossiter good-by, said that he would never forget her, and never would be friendly with his friend's family. It was not till now that Miss La Ross realized that a really good life results in an excellent impulse, but the

sequel proved the truth of Miss La Ross's predictions. One night, a year before Prince William's death, when she was wearing a diamond star in her hair, and admiring it, she heard Jimmy's front door open in the hall, and, turning round to look through the open doorway, in came Jimmy, and after that where Jimmy went Alfred would sometimes go to the house next door, and they went over this now, and ever since Miss La Ross's spirit had been there, and Jimmy had often insisted that a marriage should be accomplished in some ancient house–"And we will found a family of our own," said Jimmy, "a for some distant house in this great Belgium."

The days went on and it was a glorious spring morning. Early in March a thick autumn rain had fallen on the house in New York and the leafless trees looked weird and strange, and the wind roared like a crested wave of fire. In the afternoon the thunder roared and soon an occasional gray drizzle eddied about the wet streets and the city glow and tint began to merge into a world of pink and red and amber and purple and sunset colors.

The low ceiling pillars had covered their treasures of gold and gems, and the great stained-glass windows in dreamily gentle, half-lights, had arranged themselves in red and gold stripes of magic tints. Even though the delicate color scheme had had its disappointments once, it had certainly affected one hour in this room as well as ten months later and longer stays at least would have done.

To-day a soft South wind had drifted through the open door, and a sudden east wind had driven open the French windows of Miss La Ross's bedroom, and it rained in pure flames between the ceiling and boarded floor. Alfred's room was fragrant with his presence.

> "... A little singing bird
> That, living in a cage, demands a friend
> Whose cool-blooded lure can warm the heart with love
> To a fluttering, wounded thing.

As in a pathetic garden, so in the hall room.

It was cold, to-day. Already the rooms seemed overheated. The curtains were already half drawn.

She shivered.

"Mid-winter, to-day," thought Alfred, watching the sweep of Ann's white shoulder and patting her thin cotton frock. "Seven there were of days. And seven is yet untold gone. Fine, fine day, by Christ! Come out of this old soot, and we'll fly... Away. God rest his soul from hell, if ever such a devil crawled this broad, raw earth.... Where are you, Ann?"

Ann waited and trembled, she knew not why, for a sharp voice was asking suddenly for the check book in her hand.

"Get me change enough to pay for lunch for Jimmy," Alfred chided.

Before the one empty chair on the hall table and under the curtains lay a crashing pile of ready money. "And the window shades are closed," added Alfred.

"It won't shut out the rain," smiled Ann.

"But he won't care," protested Ann.

Alfred laid a strong withdrawing hand on the fair golden hair for a moment.

"It's all right," he coaxed. "Without a cent behind them to-day we can put in four thousand and close the bottom against a falling price like this." He was holding up the window sill six inches.

While he stood she whispered:

"I'm only lucky to save the day."

"He helps you without a reward," Alfred said.

"He's kind... and darned bad."

Ann noted dangerous things that afternoon.

"You could sing and play?" she asked.

"No, no!" insisted Alfred. "I CAN'T play and sing. The room is cold. It's warm within."

Alfred was changing clothes when he had that lucky escape, and Alfred momentarily forgot his debt. Ann laid the bill she had placed on the table, and when she had gone Alfred had not even looked at it, and it was the act she saw in that frame of mind, remembering it, that made her put it back again.

Now Alfred was thoroughly cold and temperamental, and when he probed an obligation that he had just been trying to shift on the other fellow, he was more easily reminded. When Jimmy, cold and hungry, had wormed his way into his room that day at dinner, and been halted at his close chair by the soup stove, the young man's gaze had fixed furiously to the platter of gold and had immediately started on the other food with an intensity of expression that had awakened Jimmy's appreciation of the hot day of purposes and had aroused even Ann's observant sense.

Jimmy's employer had met him on Close Street after the unsuccessful row over the Dearborn Cats. Jimmy, who was not naturally an observant boy, had tried to keep in the line of his employer's movements and tell Alfred his employer just what he did for a living, but all Alfred's energy had vanished, and on sundry occasions he had caught Jimmy's eye, and once he had promptly appeared to mere assiduous examination of the window. Employer's Jimmy had been dexterous enough, subdued, but his dexterity and subtlety and sagacity had not failed.

As one in employment was a most elusive proposition in this crafty world of facts, just then Alfred had found a perfect driftwood, and so had met and accepted and stood in the way of Jimmy's castigation and reproach. That is to say, he had saved Jimmy from seeing any of his own real qualities, and the critics, he had been asked in Jimmy's more frequent matinees to erase Alfred's sneer and snip off his coat, and he had instantly become a mental picture of Jimmy Dean's assistant to the lawyer and the college professor.

It was Jimmy's reckless impetuousness, not his single fearless single energy, that had led Ann through the door at sight of Ann, that had

electrified the tremendous audience, not her own act or attitude. Jimmy had thought still of the boy as a fellow mortal, now his master had gone.

That was a satisfactory driftwood, of Jimmy.

That evening Ann's maid had gone into the bedroom where Jimmy was and had said again that he looked very tired, and as Ann assigned as a reason his long face, it was not impossible to conclude that he was suffering from a formal ache.

But, as she had only managed a nap and had been rather silent when away from Jimmy, Ann determined that the tired-out and worn-looking man must be tired, too, and she secretly determined that at breakfast to-morrow she would place in his hands all the evidence she could imagine against Jimmy, as pertaining to the unsolved mystery of Jimmy Dean's friend.

CHAPTER II

That afternoon Jimmy was taken into the professor's home to dinner. Wonderful Dean's Hal

**The Diary of Samuel Pepys**

Context (826 bytes):

June 1st. Having taken our leaves of Sir W. Batten and my Lady, who are gone this morning to keep their Whitsuntide, Sir W. Pen and I and Mr. Gauden by water to Woolwich, and there went from ship to ship to give order for and take notice of their forwardness to go forth, and then to Deptford and did the like, having dined at Woolwich with Captain Poole at the tavern there. From Deptford we walked to Redriffe, calling at the half-way house, and there come into a room where there was infinite of new cakes placed that are made against Whitsuntide, and there we were very merry. By water home, and there did businesses of the office. Among others got my Lord's imprest of L1000 and Mr. Creed's of L10,000 against this voyage their bills signed. Having wrote letters into the country and read some things I went to bed.

MambaByte-972M:

2nd. Up betimes and to my office, where we sat all the morning, and at noon to the 'Change, where I met with Mr. Castle, who told me how the Commissioners of accounts for the time past are so many and so ill as we could not tell what to do to make them all alike to a safe composition, and therefore prayed me to come to a meeting of the East India Company, which I did, and there I found Sir W. Batten, and Sir J. Minnes, and Sir John Chichly, and Mr. Castle, and there I did give them a full and large answer to many things objected against them, and so went away, and then Sir W. Batten and I to the Coffee-house, where we did not speak to any body, but sat talking till 9 at night, and so home and to bed.

3rd. Up betimes and to my office, where all the morning very busy about my accounts, and so at noon to dinner, and thence to the office all the afternoon, and in the evening by coach to St. James's, and there find Sir W. Coventry, and I with him and Secretary Morrice into the garden to the house, where we did very little business but stood below in the entry and

heard the great dispute between the two recorders in Chancery, my Lord
Chief Justice Keeling and Mr. de Lyne, who is a cunning fellow and very
much craft in his humour. But the judges pretty well where they are, and
so did discourse and talk together of heat till supper, and after supper
did discourse together about the same matter, and so to bed.

4th. Up betimes, and to the office, where all the morning, and at noon
home to dinner, and then to the office again, and there all the afternoon
till night, and then home to my wife, and after supper to bed, my eyes
being bad.

5th. Up betimes, and to my office, where the greatest part of the
morning about my accounts, and so at noon to dinner, and then to the
office again, and there till the evening, and then home to my accounts,
and then to my office again, and there late doing business, and so home
to supper and to bed. This day I hear that Sir W. Pen is come unexpected
from Portsmouth to towne, after being absence all these years.

6th (Lord's day). Up betimes, and an hour with my wife in her chamber
about putting things and other right as to our house to my account, which
I do, and then with her to church, and then home, and dined with my wife,
and so to walk a little in the garden, and then to my office, and there
down by water to Deptford, where we did not go on the wall as I have been
accustomed, and there took in Shish

    [Shish, an old oarsman. A short coarse fishing-net, such as the
    Irish carry their fish on.]

boat coming after us, and so landed him and walked home, and I landed at
Greenwich, and so to the office, and there sat all the afternoon, and did
much business, and so home to my wife, who was come home with great cold
late, having gone this day with her father and mother wager to Mrs. Browne
to the Bath, and after supper to bed, my eyes being bad.

7th. Up betimes, and to my office, where to our morning's work again,
and then home to dinner, and after dinner abroad with my wife to the
New Exchange, by the way calling at Mr. Harper's, at whice, seeing the
fashion, I went in, and there did give her my French grammar, which she
likes well, and so to the Cross Keys at the Strand end, and there drank
and parted, and I to my Lord Crew's, and there dined with him, and had a
good dinner, and good company; among others, my good friend Mr. Howe, with
whom very good discourse all dinner time. After dinner to the office,
and there till late both with Captain Taylor and Harman and I did draw up
some new things for them in order to our consultation tomorrow, and so
home and to bed.

8th. Up betimes, and to my office, and thence by appointment to the
Exchequer about finishing my account of Sir J. Minnes' demand of his
certificate touching his late imprisonment at Tangier, which I did, and
mightily accounted by every body, and after signed and so to and again
with Sir W. Warren, who dined with me, and was very well pleased with my
account a' I have done in it, and so away, and I to the office, where we
sat all the morning, and at noon dined alone with Sir W. Batten, which I
have not done a great while, but I believe I shall 'impulsas pietatem
tuam'. After dinner to the office again, and then staid late at my
office, and so home to supper and to bed. This afternoon Sir J. Minnes
sent to me to come to him to Sir R. Ford's, whither I by and by went, and
there he and I waited all the afternoon talking together about the Fleete
and the newes we have of the present unready to be got up fit to serve the

King, and now built and ready to go to sea, and would fain have some of the King's ships fit to go to sea with the yeare almost. At last we broke up and I away to Sir R. Ford, and there sat and talked about an ancient volume of newes out of the Harleian Collection, wherein the imposthume is, and the manner of it, and the custome of putting it up and down in the cabinets of people that are sicke, which is very good. He told me the whole occasion of it, and how it was read before the Duke of Yorke and many of the Commissioners of the Navy, and how he being there was called to the King and did give and gave them his Royall thanks for their care and diligence in the care of the Navy, and that he did enlarge to them of the services and commands then expected from the King, that he would have bestowed upon them (notwithstanding the indignity which he did them) had they proceeded against him in the Navy Board, which he did move in the business of the prizes. He told me that it is believed that the King will not have the Duke of Yorke to succeed him in the command in the Narrowgan, which he and his commanders did intend to have done, and that he did die with a discontent that he must be called to the Council of State, and that there is no intention of his having that charge any more, as they told his Majesty he hath done. He told me that the Duke of Buckingham is dead, and that there is another great man, they call Hollis,

> [Thomas Hollis, a native of Germany, who had been a famous swordsman in the reign of Charles I., and afterwards one of the Commissioners of the Admiralty. He was knighted and received a pension of L500 per annum for life from the Crown. He died in 1669, aged 80.]

which is very strange, but I fear the child will be too much for him. He tells me he is certain that the Queene hath married somebody, and that he is sure that the King will never know who it was, for he did take notice to him that the Duke of Yorke did speak well of her to him, and that he did say that he would look after her, but that he was a man at all taken with the King's beauty, and did see her little short thick parts, and the motions of her eyes, and that he did not think she was a handsome woman. He says that the King did tell him that he did think that there was some command laid upon the ships that they should not go about or stay for the Queene, that he would not take that upon to the Navy

**The Patrol of the Sun Dance Trail by Ralph Connor**

Context (1, 059 bytes):

> High up on the hillside in the midst of a rugged group of jack pines the Union Jack shook out its folds gallantly in the breeze that swept down the Kicking Horse Pass. That gallant flag marked the headquarters of Superintendent Strong, of the North West Mounted Police, whose special duty it was to preserve law and order along the construction line of the Canadian Pacific Railway Company, now pushed west some scores of miles.
>
> Along the tote-road, which ran parallel to the steel, a man, dark of skin, slight but wiry, came running, his hard panting, his streaming face, his open mouth proclaiming his exhaustion. At a little trail that led to the left he paused, noted its course toward the flaunting flag, turned into it, then struggled up the rocky hillside till he came to the wooden shack, with a deep porch running round it, and surrounded by a rustic fence which enclosed a garden whose neatness illustrated a characteristic of the British soldier. The runner passed in through the gate and up the little gravel walk and began to ascend the steps.

MambaByte-972M:

Indescribable and yet unconscious was this man. All his being rose to a sense of the security that was about him as he walked onwards swiftly between rows of young firs of Canadian growth. He swung impulsively by the gap of a gate and paused. Then the clangour of an alarum bell from a city clock was sharply repeated with its deep and terrible sound.

This bell not only announced the desire of the inhabitants of the intervening concession lines to their work, but it also warned from the further side of the high boundary constructed by the late engineer of the West Mounted Police the abrupt and rapid career of another man. The pistol had fallen, and the runner was searching for another.

The solicitor buried his face in his hands.

"In God's name, man!" he ejaculated. "How do I know? Haven't I just found one?"

He had rung the bell, and now fell back among the shadows and the rough bushes. The sound startled him. He stood transfixed a moment before he looked about him.

"There's only one man knowing where that tote-road goes." He still held his breath and then listened. "'Tis where the first news comes from, and–oh, I see! There's my boy."

He waited a moment. Then once more he passed along the gap and descended the rocky little to the more sheltered spot where the new buildings stood.

He was an under officer in that high branch of the service known as the Union League, and a short sharp beat was his upon the white drilling of bashi-bazouks. With the sounding of another great bell of a church close at hand he moved quickly round to the other side of the buildings. As he approached, however, he took from his pocket a thin black silk neckerchief. It was damp and stained with the blood of dead men. He laid it in the hands of a slim girl, with the limpid blue eyes of the Canadian Saskatchewan.

"What's that for?" he demanded.

She looked as if there had been something she desired to say, then left the agitated conclusion unfinished. Her eyes sought his in the pathetic wistfulness of a child, then suddenly fell. For the hurt he had done her was not a wound incurred in battle. It was merely a little scratch in the hand, and let alone that, in a manner of speaking, it was all she had. The blood of a man is always more significant than that of a scratch on the bark of a tree, and a pressure of the earth leaves a deeper mark on a man's arm. With a sigh the runner removed the blood stain and turned his face towards the sound again. He walked half across the open grass from which he had sprung. From his ample form to the far-distant leaping folds of his drilling trousers he had trailed a forked stick, and so to the girl.

In a few seconds he came back.

"It's me, pardner, Superintendent Strong. It's me I'm goin' down from the Soo, for the job I had in Mexico after I came out here. I'm connected with the Canadian Pacific Railway and they're hunting up a man who did have a finger wounded by a Canadian rock. I'm sendin' the little flag

with her." He emphasised the word "flag." A rough skin mark, furrowed in a straight line down his left cheek, marked the place of the scar and brought him to a sudden stop. His eyes were on the scrolled letters above his head.

"I'm going down to get it. I've got to get it to the bottom, anyway, for divil a bit of paper they'll let me have at British Columbia. Oh, God!"

He raised his voice. In a moment he had departed. In a few minutes he had rejoined the girl. They rejoined the solicitor and returned with him to an open space before the meeting place of the railway company. As they gathered round a table spread with an untasted meal the solicitor spoke. The railroad company was working out from British Columbia to Montreal.

"In our fight we had it hard," he said. "The northern route to League Island was blocked, we could not reach there to recruit. We had to look for a northern route, for there was none. At first the league flag of Ottawa was given up. That was only till October. Then a young man on the ground from London came to us. He'd been in the runner's service along the whole line from Montreal. He was headed for Canada on the telegraph. Two of us had to flag him as soon as we set out from here. He had been over that ground about fifty times before, and knew the whole road well for forty miles. The head of us did not know it till he came to the junction where the main line crosses the north line of the United States. We took that name on the tin to test him."

"What was the corporation over there for?" said the solicitor. "I remember, I remember. It occupied a part of the big Kelvin mine. I was helping get the first claim post run by the Union League at the time I was there. He was out hunting coal. He came down one day to see the coal pits about the ground. On the way he was stopped and accused of raising a rebellion, and was arrested and taken to the Soo, where he was made to give evidence in a certain case that had been laid before him."

"And what was the precise cause of the complaint?" asked the runner.

"Well, it wasn't a case at all, it was a fact. That's all," explained the constable.

"From what I heard then of the runners of the London and North West, their work wasn't near so exciting and dangerous as it had been reported to be. Also it was the work of others, others still, and they were arrested. They was a young feller and a girl married over two years ago, and he was shot."

"Brought to trial for that by himself or his relatives or some of the men who were with him?" There was a puzzled, gentle expression on the face of the railway superintendent. He was of much higher rank, for he had not been present at the trial of the accused. He glanced up at the runner.

"Arrested?" The bit of food in his mouth was working like a millstone in the Soo employer's breast. Then, as though unconsciously to himself, his lips said "yes" instead of "no," and he added instead, "and sworn to it. That's as far as you've got, pardner. Anything else, sir?" He was watching the silent figure with intense desire to see his face and to know what he felt. It did not come, and he settled himself in his chair with a sigh.

"That was short work. They marched the young feller up here, and give him the Canadian division. It was the station sergeant-inspector from the

Canadian line sending down from headquarters to show he was all right and not having heard anything against him. And if you don't know that it's not the worst of the testimony we have to give, pardner. It wasn't the best. The fact is the young man was getting three weeks' sentence at the time."

"That was only a month ago," broke in the businesslike runner, who had been preparing himself for a full report. "What had he done? Tell us?"

