# OpenReview forum: "MambaByte: Token-free Selective State Space Model"
_colmweb.org/COLM/2024/Conference — COLM_

### Official Review · Reviewer_jqQ8 · 2024-05-09

**Rating:** 8
**Confidence:** 3
**Ethics Flag:** 1

**Summary:**

This paper proposes MambaByte, a byte-level language model without tokenizers. Compared with subword tokenization, byte-level language models are more robust and have fewer inductive biases. However, the resultant sequences are much longer and bring challenges in efficiency, specifically with Transformers. For efficient training, the authors build MambaByte upon Mamba architecture. For efficient decoding, a variant of speculative decoding is proposed. Experiments verified the efficiency and robustness of MambaByte.

**Reasons To Accept:**

1. MambaByte demonstrates promising potential towards tokenizer-free architectures. Looking forward to seeing MambaByte as a larger-scale model and operating for various downstream tasks.
2. Experiments are solid and comprehensive, comparing MambaByte to Transformers, SSMs, and MegaByte. Results show the superiority of MambaByte in both efficiency and robustness against both byte-level models and subword Transformers.
3. Notably, MambaByte shows impressive performance with longer sequences.

**Reasons To Reject:**

I do not find any clear reason to reject this paper.

---

> ### Author Rebuttal · Authors · 2024-05-30
>
> We thank the reviewer for their confidence in our work.

---

### Official Review · Reviewer_Nr6r · 2024-05-10

**Rating:** 7
**Confidence:** 2
**Ethics Flag:** 1

**Summary:**

This paper proposes MambaByte, a byte-level Mamba state space models and observed that Mamba efficiently addresses issues in byte-level language modeling without the need for specialized architectures like global patching. Experiments on language modeling tasks shows MambaByte demonstrates competitive performance both on language modeling performance and efficiency.

**Reasons To Accept:**

- The proposed token-free adaptation of Mamba, MambaByte, demonstrates competitive performance with subword Transformers and maintains the benefits of token-free
language models
- This paper also proposes an adaptation of speculative decoding for byte-level models (subword model for drafting, byte-level verification), which shows a 2.6× inference speedup
- MambaByte shows great performance for length extrapolation

**Reasons To Reject:**

I have noticed that different scales of MambaByte are used in various comparisons. Could you provide a comprehensive comparison of MambaByte at different scales? Also, how does scaling affect different models, and will MambaByte demonstrate improved scaling factors?

---

> ### Author Rebuttal · Authors · 2024-05-30
>
> We thank the reviewer for their meaningful comment on improving our paper.
>
> We employed different scales of MambaByte across the paper to enable iso-FLOP and iso-parameter comparisons with MegaByte. We agree that this makes it hard to see the scaling curves. To facilitate easy comparison, here are all the results in one table (all models are evaluated using a context size of 2,048 tokens):
>
> | Parameters |  | Effective tokens trained | Test PPL ↓ |  |
> |---|---|---|---|---|
> | *Mamba* | *MambaByte* |  | *Mamba* | *MambaByte* |
> | 352M (d=1024, n=48) | 353M  (d=1024, n=53) | 60B | **43.0** | 44.4 |
> | 578M  (d=1328, n=48) | 581M (d=1792, n=48) | 80B | 39.5 | **39.4** |
> | 1.03B  (d=1792, n=48) | 972M (d=2304, n=48) | 150B | 36.7 | **35.9** |
>
> To answer your second question, these numbers indicate that the model scales similarly in parameters to Mamba (see Section 4.3.1 in the [Mamba paper](https://arxiv.org/abs/2312.00752)), which may be because both models have nearly the same hidden state size. We will update our findings regarding the scaling laws in the final version of the paper.

---

### Official Review · Reviewer_m6Tj · 2024-05-13

**Rating:** 5
**Confidence:** 4
**Ethics Flag:** 1

**Summary:**

Authors propose using state space models to learn a language model on byte sequences utilizing the fact these new family of models have an internal memory that is independent of the context window length.


The authors frame the paper as an improvement of byte modeling with other byte models as the baseline. I believe the right baselines are the SOTA LMs regardless of tokenization. In this case, we benefit little by showing one byte model better than the other. We benefit more by understanding how much the gap remaining from the wordpiece/bpe models is.

**Questions To Authors:**

(1) Please, add (Training/inference) isoFLOP analysis of MumbaByte vs Mumba models to prove the computational advantage of your approach.

**Reasons To Accept:**

(1) eliminating the last preprocessing/postprocessing step in language modeling has huge practical implications in simplifying systems and reducing sources of errors.

(2) Pushing modeling to support small vocabulary models allow us to extend the lessons learned in language modeling to genomics and image modeling efficiently.

**Reasons To Reject:**

(1) The authors compare Mumba (Wordpiece) vs Mumba Byte in terms of equal params (table 3) but do not have a baseline of the wordpeice based models in terms of training compute match. I would like to see figure adjusted with the curve that correspond to Mumba word piece model to figure out if it is more efficient than MumbaByte (training wise).


(2) Adaptive decoding seems a general technique that could be also used to speed up any language model model regardless of the tokenization. I would like to see the speedup achieved for MumbaByte vs Mumba both applying adaptive decoding.

(3) The way that adaptive decoding still rely heavily on word-piece level models kind of negate the benefits that we are seeking by eliminating the tokenization.

(4) Missing references to relevant papers in the field:
(a) https://arxiv.org/pdf/1808.04444
(b) https://arxiv.org/abs/1908.10322

---

> ### Author Rebuttal · Authors · 2024-05-30
>
> We thank the reviewer for their critical comments.
>
> (1) We agree with the reviewer that comparing MambaByte to Mamba in Table 3 does not facilitate iso-FLOP comparison, and training might favor wordpiece models.
>
> Since the submission, we were able to continue training Mamba-1.03B for the full 600B bytes (4x more than MambaByte, thus matching the total training compute used for MambaByte-972M) and note the (test) perplexity to be 33.9. Additionally, we observe that Mamba achieves near-optimal performance more efficiently than MambaByte, but not 4x (as expected) and more so, 2.2x. We also observe that the perplexity for Mamba-1.03B does not improve significantly beyond 150B training bytes, which aligns with the observations made by [Rae et al. (2019)](https://arxiv.org/pdf/1911.05507).
>
> We will update the iso-FLOP curves in the final version of the paper.
>
> (2) We agree that speculative decoding can also speed up wordpiece models. However, we argue that byte-level models have much lower entropy than wordpiece models and can therefore be sped up significantly using this approach, more so than wordpiece models. At best, Transformers can be sped up 2x with this method, and the subword Mamba is already significantly fast to begin with. With a carefully-engineered multistep hardware-aware kernel (from a parallel work), we observe a 1.7x speedup for subword Mamba with speculative decoding (which is lower than the 2.6x speedup achieved for MambaByte).
>
> (3) We disagree with this argument. While the speculated samples are still from a (smaller) subword model, they have all the benefits of byte-level models (robustness to noise, unbiased by tokenization, etc.). The smaller wordpiece model is only used to facilitate faster inference. While this can bias sampling, the impact was observed to be minimal. We present evidence of this in Table 6, where we show that our speculative decoding using subword drafting produces text more similar to MambaByte-972M than Mamba-1.03B, both of which have different inductive biases.
>
> (4) We will include these references in the final version.

---

### Official Review · Reviewer_13GY · 2024-05-14

**Rating:** 8
**Confidence:** 3
**Ethics Flag:** 1

**Summary:**

The paper presents a token-free selective SSM called MambaByte, which is for improving both modeling performance and computational efficiency. In addition to building upon Mamba architecture, it integrates speculative decoding for subword drafting and byte-level verification.  The paper conducts experiments from compute-matched and parameter-matched perspectives to demonstrate the effectiveness and efficiency of the proposed method. The main comparisons are made between MambaByte and Transformers, SSMs, and MegaByte architectures. The results show that MambaByte outperforms the compared byte-level models in several datasets.

**Questions To Authors:**

1. Mamba architecture has been questioned for how it can be applied to downstream tasks and how it performs on CV/NLP domains. I wonder if the authors have tried the proposed model on downstream tasks to compare the performance with traditional architectures.

**Reasons To Accept:**

1. The motivation is clear and easy to understand. The presentation and writing are of high quality.
2. The idea of using Mamba as the draft model and MambaByte as the verification model is interesting.
3. The experiments are quite extensive to show the superior performance of the proposed model. The parameter-matched and compute-matched evaluations are fair and able to provide more observations into the performance among different architectures.

**Reasons To Reject:**

I don't observe very obvious reasons for rejection. This is a solid paper in general.

---

> ### Author Rebuttal · Authors · 2024-05-30
>
> We thank the reviewer for their confidence in our work.
>
> We acknowledge that recent studies have shown that, at very large scales, Mamba seems to be worse than Transformers on in-context learning (ICL) tasks. While we hope to compare MambaByte to MegaByte on similar downstream tasks, the models employed in this work are too small for these benchmarks. Furthermore, from past works, we believe that perplexity strongly correlates with downstream performance and  mainly focus on achieving a lower perplexity.

---

> > ### Comment · Reviewer_13GY · 2024-06-02
> > **Thanks for the response**
> >
> > Thanks to the authors for their response. After reading other reviews, I still hold positive opinions on the submission and maintain the same score. I think it might be influential for future directions.

---

### Decision · Program_Chairs · 2024-07-10

**Decision:**

Accept

**Comment:**

The paper proposes to a method (MambaByte) to train byte-level language models based on the Mamba architecture. Due to the linear (instead of quadratic) scaling with respect to sequence length, MambaByte is more efficient than Transformer byte-LMs and achieve better language modeling performance. The paper proposes a speculative decoding technique to use a sub-word Mamba LM as draft model to speed up MambaByte decoding by 2.6x, overcoming the low inference issue of byte-level LM.
+ The evaluation is careful and comprehensive. MambaByte is compared to models with the same params as well as same training
budget.
+ The speculative decoding method with a sub-word model as a draft model is clever and elegant.
+ The resulting byte-LM is more robust to character-level noise or corruption.
- The paper could benefit from discussion of scaling to larger LMs (in cases where the weight matmul takes proportionally more time and attention / SSMs do not take as much time).
Overall the paper is a valuable contribution to the direction of token-free language models.